# WINDOW-BASED HIERARCHICAL DYNAMIC ATTENTION FOR LEARNED IMAGE COMPRESSION

## ABSTRACT

Transformers have been successfully applied to learned image compression (LIC). In fact, dense self-attention is difficult to ignore contextual information that degrades the entropy estimations. To overcome this challenging problem, we incorporate dynamic attention in LIC for the first time. The window-based dynamic attention (WDA) module is proposed to adaptively tune attention based on the entropy distribution by sparsifying the attention matrix. Additionally, the WDA module is embedded into encoder and decoder transformation layers to refine attention in multi-scales, hierarchically extracting compact latent representations. Similarly, we propose the dynamic-reference entropy model (DREM) to adaptively select context information. This decreases the difficulty of entropy estimation by leveraging the relevant subset of decoded symbols, achieving an accurate entropy model. To the best of our knowledge, this is the first work employing dynamic attention for LIC. Extensive experiments demonstrate the proposed method outperforms the state-of-the-art LIC methods.

## 1 INTRODUCTION

Vision Transformer (ViT) has achieved tremendous advancements in the field of computer vision, with many studies applying it to learned image compression (LIC) methods (Zhu et al., 2022; Qian et al., 2022a;b; Liu et al., 2023). Efficient self-attention between all sequence elements helps the model pay attention to long-range information. There are mainly two aspects of works transferring CNN-based learned image compressions to ViT architectures. Utilizing Swin Transformers (Swin-T) (Liu et al., 2021b) in main encoder-decoders to build powerful nonlinear transforms (Liu et al., 2023; Zhu et al., 2022; Zou et al., 2022; Lu et al., 2022). On the other hand, some works leverage ViTs in entropy models to capture global contextual information, supporting a more accurate probability estimation of the latent representation distributions (Qian et al., 2022a; Koyuncu et al., 2022; Kim et al., 2022). However, the rate-distortion (RD) performance improvements of these methods are marginal.

Nonlinear transformations and entropy models are the key components of LIC. Despite ViTs enables attention to more distant context, it does not guarantee compact transformations and accurate entropy estimation. Adjacent features exhibit stronger causal relationships and the previous work (Minnen et al., 2018) reveals that convolution kernel sizes larger than $5 \times 5$ (with larger receptive fields) unexpectedly compromise the RD performance. The works (He et al., 2021; Zou et al., 2022) also prove that redundancy primarily exists in local regions. Some redundancy information indeed exists in distant regions (Qian et al., 2022b), but referring global contextual information increases the risk of overfitting. Previous works have focused solely on the long-range modeling capabilities of ViTs, ignoring the issue of overfitting.

In this paper, we analyze the challenge of applying ViTs to image compression, and a novel method is proposed: **dynamically sparsifying attention.** Plain Swin-T compute paired attention between all elements in local windows. In other works, all decoded symbols in a window are considered when decoding the current node. Different from recognition tasks, the goal of compression is to remove redundancy. Reference to irrelevant content can mislead probability estimations. We claim that the core contradiction of overfitting is that the attention-pattern space of ViTs is much large than redundancy-pattern space. To overcome this problem, we sparsify the attention-pattern space and propose a compression model adopts adaptive attention patterns learning from the entropy distribu-

tion of the image. Specifically, the model is built on the window-based dynamic attention (WDA) module, which tunes the attention matrix to ignore useless references in local windows. The WDA module works in multiple feature scales to hierarchically tune long-range attention patterns. Furthermore, the dynamic-reference entropy model (DREM) is proposed, which builds upon the concept of dynamic attention patterns, adaptively selecting reference contextual information for the current encoding element based on the known entropy distribution. The aggregation of relevant decoded symbol subsets significantly reduces the difficulty of probability estimation in the entropy model.

To the best of our knowledge, we first focus on the overfitting issue of transformer-based learned image compression methods and modulate the attention by the means of dynamic sparsification patterns. In summary, our contributions can be concluded as follows:

- We first integrate dynamic attention into learned image compression, which narrows the gap of attention space and redundancy space. Sparse attention patterns ignore globally irrelevant contexts, reducing the risk of overfitting.

- We propose the window-based dynamic attention (WDA) module, which adaptively learns attention patterns from entropy information of latent representations. The WDA modulates the attention matrix at different scales in a fine-to-coarse manner.

- We present the dynamic-reference entropy model (DREM) to select subsets of decoded symbols, which provide enough contextual information and simultaneously reduce the optimization difficulty.

- Experiment results demonstrate that our proposed method achieves 13.42%, 17.74% and 12.93% BD-rate gains over VTM-17.0 on the Kodak, Tecnick and CLIC datasets respectively and outperforms the state-of-the-art LIC method MLIC++.

## 2 RELATED WORK

### 2.1 LEARNED IMAGE COMPRESSION

Early LIC methods adopt convolutional neural networks (CNNs) in both encoder-decoders and entropy models (Ballé et al., 2018; Minnen et al., 2018; Lee et al., 2018). (Cheng et al., 2020) first incorporates the attention mechanism into LIC, which pays more attention to regions with complicated textures. However, the local receptive field of CNNs limits their ability to capture long-range spatial dependencies. Some global methods utilize non-local networks (Chen et al., 2021) and content-weighted attention masks (Li et al., 2018; Mentzer et al., 2018) to alloacte bits across the entire image, leading to an overall improvement in RD performance. With the rise of transformers, ViTs are gradually emerging in LIC. The global self-attention constructs more powerful nonlinear transformations (Lu et al., 2022; Zhu et al., 2022; Liu et al., 2023; Zou et al., 2022) and provide rich contextual information in entropy models (Qian et al., 2022a; Kim et al., 2022; Liu et al., 2023). However, the downside of global information is that irrelevant context increases the difficulty of entropy estimation and the risk of overfitting. We propose the WDA module to dynamically sparsify the attention patterns to address the problem.

### 2.2 DYNAMIC ATTENTION

Previous works have demonstrated that a significant amount of computational redundancy exists in ViTs. Only a small proportion of tokens contribute to the final prediction, thus removing those useless tokens improves the computational efficiency without harming the performance (Chen et al., 2023; Wei et al., 2023). Following that, some sparse attention methods are proposed to accelerate ViTs, including token sampling (Rao et al., 2021; Fayyaz et al., 2022; Tang et al., 2022) and attention masking (Liu et al., 2021a; Kitaev et al., 2019). Among those methods, static sparse methods (Tay et al., 2020; Kong et al., 2022) introduce heuristic sparse attention patterns with challenging of generalization. While dynamic methods (Yin et al., 2022; Venkataramanan et al., 2023; Lee et al., 2024) learn dynamic attention patterns from data in a flexible way. Our work is inspired by dynamic sparse attention but applied in a different domain. Specifically, we apply the dynamic sparse attention to more efficiently eliminate representation redundancy rather than to reduce computational complexity.

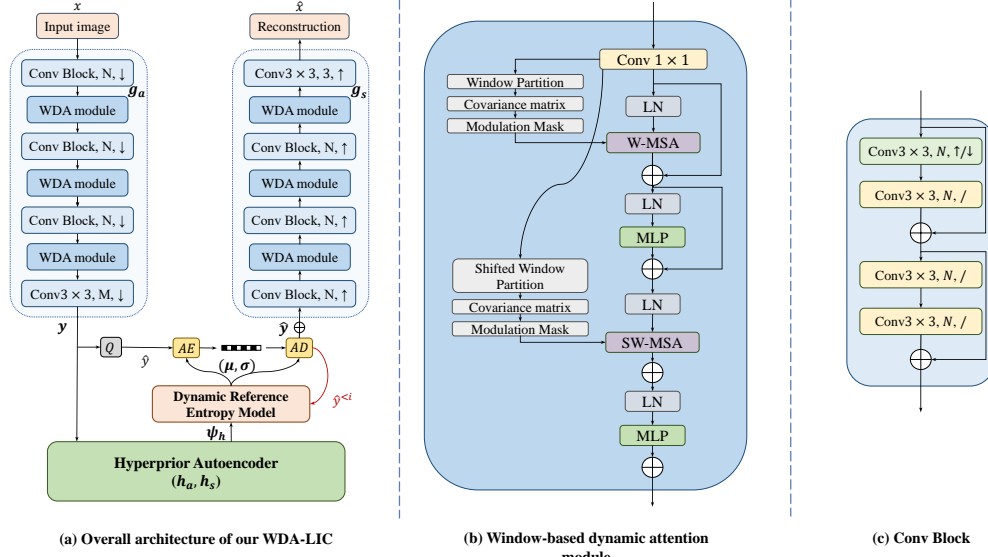

(a) Overall architecture of our WDA-LIC  (b) Window-based dynamic attention module  (c) Conv Block

Figure 1: Overall framework of the proposed **W**indow-based **D**ynamic **A**ttention **L**earned **I**mage **C**ompression (WDA-LIC). $g_a$ and $g_s$ denote analysis and synthesis transforms consist of multiple Conv Blocks and WDA modules. Conv Blocks extract local features and the WDA modules capture long-range contextual information with dynamic attention patterns. $N$ denotes output channel numbers in every layers.

## 2.3 CONTEXT ENTROPY MODELING

In LIC entropy models replace the marginal probability distribution of latent variables by the joint probability distribution with prior variables to reduce entropy (Ballé et al., 2018; Minnen et al., 2018; Lee et al., 2018). Due to the sequential property of decoding, leveraging previously decoded features (*i.e.*, context) to provide predictive information for the current decoding step can significantly reduce the joint entropy. And an optimal contextual pattern determines the upper bound of prediction accuracy. Some works divide feature channels into multiple slices and remove redundancy by leveraging correlations between channels (Minnen & Singh, 2020; He et al., 2022; Zhu et al., 2022). In terms of spatial redundancy, CNN-based methods capture local correlations between neighboring representations (He et al., 2021; Zou et al., 2022; Guo et al., 2021) and transformer-based methods calculate relevant information over longer ranges (Liu et al., 2023; Lu et al., 2022). Although relevant information may exist in global regions, previous works (He et al., 2021; Minnen et al., 2018) show that adjacent pixels are likely to have a stronger causal relationship. Focusing on too much irrelevant information increases the difficulty of prediction. Some methods select top-K elements in global regions to centralize attention (Qian et al., 2022a;b; Ma et al., 2021). However, this fixed-number reference pattern fails to adapt to sample differences. We propose the dynamic-reference entropy model (DREM) to adaptively select reference subsets with entropy information.

## 3 METHODS

### 3.1 PROBLEM FORMULATION

The architecture of our proposed **W**indow-based **H**ierarchical **D**ynamic **A**ttention **L**earned **I**mage **C**ompression (WDA-LIC) is shown in Figure 11. The overall algorithmic can be formulated as follows:

$$y = g_a(x; \theta_{g_a}), \hat{y} = Q(y), \hat{x} = g_s(\hat{y}; \theta_{g_s}), \tag{1}$$

where the encoder $g_a$ with parameters $\theta_{g_a}$ transforms the input image $x$ to latent represetation $y$. Following that $y$ is quantizied to $\hat{y}$, which is modeled as a single Gaussian distribution with estimated parameters $(\mu, \sigma)$ to be entropy encoded. The decoder $g_s$ with parameters $\theta_{g_s}$ utilizes $\hat{y}$ to reconstruct $\hat{x}$. It is so critical to accurately estimate the distribution parameters $\mu$ and $\sigma$. We adopt

the hyperprior model (Ballé et al., 2018) and the channel-wise autoregression entropy model (Minnen & Singh, 2020; He et al., 2022) to estimate the Gaussian parameters $(\mu, \sigma)$. The hyperprior model is used to capture side information:

$$z = h_a(y; \phi_{h_a}), \hat{z} = Q(z), \psi_h = h_s(\hat{z}; \phi_{g_s}), \tag{2}$$

where $\psi_h$ denotes the side information provided by the hyperprior model. A mount of redundancy exists between channels and the decoding process is sequential, we follow provious works (He et al., 2022; Jiang et al., 2023) to divided latent variables $y$ into $S$ slices $\{y^0, y^1, \ldots, y^{s-1}\}$ so that encoded slices provide contextual information to help the entropy estimation of currently encoding slice as shown in Figure 3. During the process of encoding slice $y_i$, all its front slices $\{\hat{y}^0, \hat{y}^1, \ldots, \hat{y}^{i-1}\}$ and the side information $\psi_h$ are fed into the proposed Dynamic-Reference Entropy Model (DREM) to estimate the Gaussian parameter of current slice as follows:

$$\begin{aligned}\Phi_i &= e(\psi_h, \hat{y}^{<i}, y^i) \\ &= (\mu_i, \sigma_i),\end{aligned} \tag{3}$$

where $e$ is the DREM. Therefore, the probability of current slice is considered as follows:

$$p_{\hat{y}^i|\hat{z},\hat{y}^{<i}}(\hat{y}^i \mid \hat{z}, \hat{y}^{<i}) \sim \mathcal{N}(\mu^i, \sigma^i), \tag{4}$$

Since the entropy bottleneck $\Psi$ is used to encode $\hat{z}$ as $p_{\hat{z}|\Psi}(\hat{z} \mid \Psi)$, the overall rate-distortion (RD) loss function is defined as:

$$\begin{aligned}\mathcal{L} &= \mathcal{R}(\hat{y}) + \mathcal{R}(\hat{z}) + \lambda \cdot \mathcal{D}(x, \hat{x}) \\ &= \mathbf{E}[-\log_2(p_{\hat{y}|\hat{z}}(\hat{y} \mid \hat{z}))] + \mathbf{E}[-\log_2(p_{\hat{z}|\Psi}(\hat{z} \mid \Psi))] \\ &\quad + \lambda \cdot \mathcal{D}(x, \hat{x}),\end{aligned} \tag{5}$$

where $\lambda$ is a Lagrangian multiplier to control the RD tradeoff. $\mathcal{D}(x, \hat{x})$ denotes the distortion term such as Mean squared error (MSE) loss. $\mathcal{R}(\hat{y})$ and $\mathcal{R}(\hat{z})$ are the bit rates of latent representations $\hat{y}$ and $\hat{z}$.

## 3.2 DYNAMIC ATTENTION-BASED TRANSFORMATION

We build the nonlinear transformations in a CNN-Transformer mixed way, as shown in Figure . At each stage, the Conv Block extracts features through a CNN, followed by a Swin-T based WDA module to fuse features within a local window. Specifically, the WDA module adopts adaptive attention patterns with the instruction of entropy distribution in local windows. With smaller feature scales, the WDA module tunes the attention locations in a larger receptive field during the transforming. The following sections elaborate our proposed WDA module.

### 3.2.1 WINDOW-BASED DYNAMIC ATTENTION MODULE

The WDA module is illustrated in Figure 2 and can be viewed as a Swin-T with dynamic attention patterns. Given an input feature $\boldsymbol{X} \in \mathbb{R}^{H \times W \times C}$, the vanilla Swin-T divides $\boldsymbol{X}$ into non-overlapping partitions $[\boldsymbol{X_1}, \ldots, \boldsymbol{X_M}]$ with $K \times K$ size windows or shifted-windows and arrange them into the feature matrix, where $\boldsymbol{X_i} \in \mathbb{R}^{N \times C}, N = K \times K, 1 \leq i \leq M$ and $M = \frac{H}{K} \times \frac{W}{K}$. The multi-head self-attention is conducted within each window $\boldsymbol{X_i}$ as follows:

$$\begin{aligned}\boldsymbol{Q}, \boldsymbol{K}, \boldsymbol{V} &= \boldsymbol{X_i}\mathbf{W^Q}, \boldsymbol{X_i}\mathbf{W^K}, \boldsymbol{X_i}\mathbf{W^V}, \\ \boldsymbol{A}(\boldsymbol{X_i}) &= \mathrm{softmax}(\frac{\boldsymbol{Q} \cdot \boldsymbol{K^T}}{\sqrt{d_k}}), \\ \boldsymbol{O}(\boldsymbol{X_i}) &= \boldsymbol{A} \cdot \boldsymbol{V},\end{aligned} \tag{6}$$

where $\mathbf{W^Q}, \mathbf{W^K}, \mathbf{W^V} \in \mathbb{R}^{C \times d_k}$ are learnable parameters and $d_k$ is the intermediate feature dimension. The above formula calculates the self-attention of each token with all other tokens in the sequence and the final output is a weighted average of all tokens within the window. Obviously,

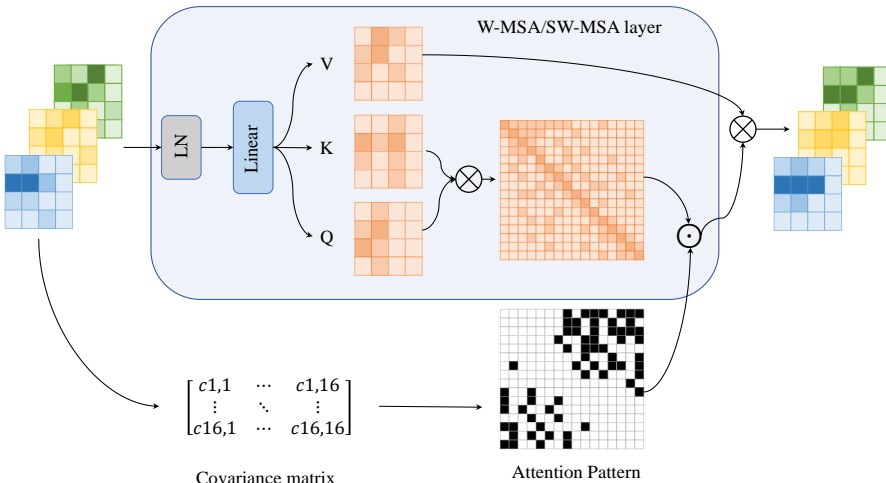

Figure 2: Proposed window-based dynamic attention module. The attention pattern is a $0-1$ matrix with the same size of attention matrix. $\otimes$ denotes the matrix multiplication operation and $\odot$ denotes the Hadamard dot multiplication operation.

spatial structure exists in attention and awful context spreads the distribution of latent features, increasing the entropy. This is not supposed to happen when coding. The correct approach is to focus attention on tokens with rich mutual information, which makes the latent features more compact and reduces entropy. Though some previous methods (Qian et al., 2022a;b) utilize Top-K scheme to select K-most relevant reference elements, this fixed attention pattern cannot adapt to the diversity of image distributions, thus the improvement of RD performance is marginal. Intuitively, regions with complex texture should reference more contextual information. Our WDA module adopt dynamic attention patterns. For each window, we can easily obtain the covariance matrix between latent variables before computing the attention matrix $V$ as follows:

$$V = \frac{1}{c-1}(X_i - \mu)(X_i - \mu)^T,$$

$$\text{with} \quad \mu = \frac{1}{C}\sum_{j=1}^{C} X_i^j, \tag{7}$$

where $V \in \mathbb{R}^{N \times N}$, $\mu$ is the mean of each token, $C$ is the number of channels. The diagonal elements of $V$ represent the variance of each variable, where larger variances correspond to higher entropy. The other elements indicate the correlation between pairs of variables. Although the covariance matrix only represents linear correlations, it is sufficient as a clue for attention aggregation. To dynamically sparsify the attention matrix, we obtain the mask matrix $M$ as follows:

$$M(i,j) = \begin{cases} 0 & \text{if } \mid \frac{V(i,j)}{V(i,i)} \mid \geq t, \\ -inf & otherwise, \end{cases} \tag{8}$$

where $t$ is the threshold, and is set to be 0.8. It is obvious that the attention pattern is dynamic due to the diversity of entropy (*i.e.*, $V(i,j)$). Following that the mask matrix modulates the attention matrix as follows:

$$\hat{A}(X_i) = \text{softmax}(\frac{Q \cdot K^T}{\sqrt{\bar{d}_k}} + M), \tag{9}$$

It is equal to adopt the Hadamard operation with $0-1$ masks shown in Figure 2. And the final output of the WDA module is as:

$$\hat{O}(X_i) = \hat{A} \cdot V, \tag{10}$$

### 3.2.2 HIERARCHICAL ATTENTION MODULATION

As the features are downsampled, the receptive field of each window corresponding to the original image gradually increases. To modulate the attention pattern in a hierarchical way, we apply the WDA module to multiple feature scales. When tuning attention at features with $d$-downsampled scales, the receptive filed to the image can be expressed as:

$$\mathcal{F} = (\frac{3}{2}K \times \frac{3}{2}K \times d)^2, \tag{11}$$

where $\mathcal{F}$ denotes the resolution of the original image $K$ is the window size. The factor $\frac{3}{2}$ is due to the shifting-window operation. Larger $K$ have a larger receptive field with more irrelevant tokens, thus the threshold $t$ tends to increase to abandon those tokens.

### 3.3 DYNAMIC-REFERENCE ENTROPY MODEL

Figure shows the pipeline of the Dynamic-Reference Entropy Model (DREM). To encode the latent slice $y^i$, all previously encoded slices $\hat{y}^{<i}$ and hyperprior context $\psi_h$ are utilized. Specifically, we split $y^i$ into two parts (*i.e.*, $y_a^i$ and $y_n^i a$ in the checkerboard spatial pattern following the previous works (He et al., 2021; Jiang et al., 2023). After coding $\hat{y}_a^i$, it provides local spatial information to $y_n^i a$. Channel-wise and global spatial context are predicted from previously encoded slices $\hat{y}^{<i}$. All context representations are concatenated in channel dimension and fed into the entropy estimation network to predict the distribution parameters $(\mu, \sigma)$.

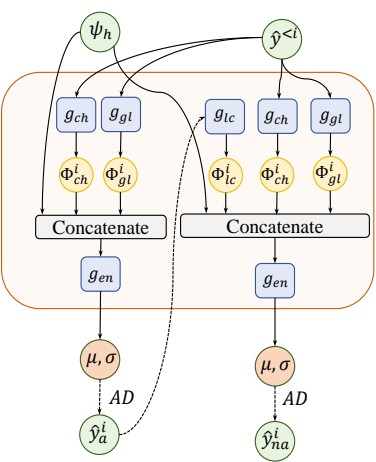

The dynamic-reference is reflected in the selection of global contextual information. As equation 8, we leverage the WDA module in the global context network to dynamically select a subset of tokens in $\hat{y}^{<i}$ according to the current entropy distribution of $y^i$. The workflow of DREM can be summarized as follows:

$$\Phi_{ch}^i = g_{ch}(\hat{y}^{<i}), \Phi_{gl}^i = g_{gl}(\hat{y}^{<i}),$$
$$\Phi_a^i = (\mu_a^i, \sigma_a^i) = g_{en}(\Phi_{ch}^i; \Phi_{gl}^i; \psi_h),$$
$$\hat{y}_a^i = AD(\Phi_a^i), \Phi_{lc}^i = g_{lc}(\hat{y}_a^i), \tag{12}$$
$$\Phi_{na}^i = (\mu_{na}^i, \sigma_{na}^i) = g_{en}(\Phi_{ch}^i; \Phi_{gl}^i; \Phi_{lc}^i; \psi_h),$$

Figure 3: Pipeline of the **D**ynamic-**R**eference **E**ntropy **M**odel (DREM). where $g_{ch}, g_{gl}, g_{lc}, g_{en}$ are networks and $\Phi_{ch}^i, \Phi_{gl}^i, \Phi_{lc}^i, \Phi_a^i, \Phi_{na}^i$ denote context latent variables. More network architecture details can be found in Appendix A.

## 4 EXPERIMENTS

### 4.1 EXPERIMENTAL SETUP

We implement the proposed compression method based on the platform CompressAI [1] (Bégaint et al., 2020). The proposed model is trained on the *Flickr*2W (Liu et al., 2020) dataset for $2M$ steps. We crop images into $256 \times 256$ patches and set batch size as 8. The Adam optimizer is utilized and the learning rate is fixed as $1e^{-4}$ for the former $1.5M$ steps. Then, the learning rate is divided by two if the validation loss hits a bottleneck (we use a wait for 20 steps). The model is trained with the RD loss in Equation 5. Following the settings of CompressAI, $\lambda$ is set as $0.0018, 0.035, 0.0067, 0.013, 0.025, 0.0483$ for MSE and $2.4, 4.58, 8.73, 16.64, 31.73, 60.5$ for MS-SSIM. For evaluation we conduct test on Kodak (Kodak, 1993), CLIC Professional Validation (Toderici et al., 2020) and Tecnick datasets (Asuni et al., 2014).

### 4.2 RATE-DISTORTION PERFORMANCE

---

[1] https://interdigitalinc.github.io/CompressAI/

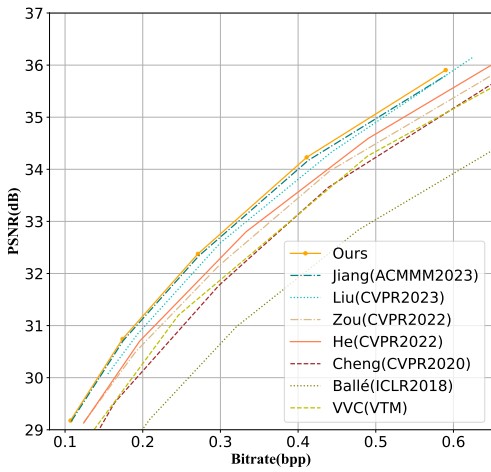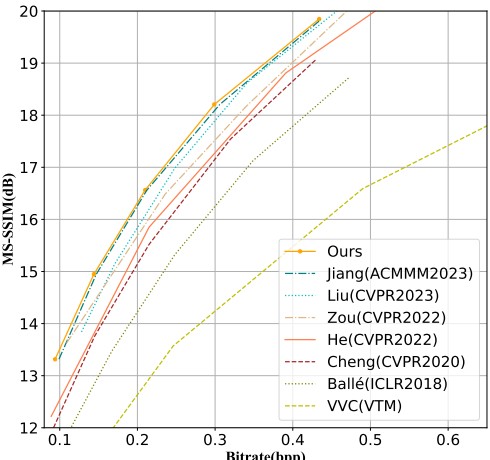

Figure 4: RD performance on the Kodak dataset (left:PSNR, right:MS-SSIM).

Six popular LIC methods (Jiang et al., 2023; Liu et al., 2023; Zou et al., 2022; He et al., 2022; Cheng et al., 2020; Ballé et al., 2018) and the traditional codec VTM-17.0 is compared. The R-D performance of the Kodak dataset is shown in Figure 4. The results of CLIC and Tecnick datasets are presented in Figure in the Appendix. To comprehensively compare the RD performance of two comression methods, we utilize the BD-Rate (Bjontegaard, 2001) metric.

### 4.3 ABLATION STUDY

**Effectiveness of the WDA module.** We remove the WDA module in analysis and synthesis transformations as the baseline and compare the BD-rate over VTM-17.0. The results are displayed in Table 2, which illustrates the efficiency of the WDA module. Atten denotes plain attention patterns that discards masks. w/ WDAtten (n=1) represents the method that maintains the last WDA module and w/ WDAtten (n=4) maintains all WDA modules. The WDA module is lightweight and is easy to be compatible with other networks. The results further shows that retaining the last WDA module still keeps performance advantage. The visualization of latent distributions are shown in Figure 5. It is obvious that the WDA module compacts the distribution of latent representations.

Table 1: BD-rate results over VTM-17.0 of state-of-the-art LICs. The evaluation is conducted on the Kodak dataset.

| Methods | PSNR | MS-SSIM |
|---|---|---|
| VTM-17.0 | - | - |
| Cheng(CVPR2020) | +5.58 | -44.21 |
| He(CVPR2022) | -5.59 | -44.60 |
| Zou(CVPR2020) | -2.48 | -47.72 |
| Liu(CVPR2023) | -10.14 | -48.94 |
| Jiang(ACMMM2023) | -13.39 | -53.63 |
| Ours | **-13.42** | **-53.96** |

Table 2: BD-rate results over VTM-17.0 on the CLIC Valid dataset of different models.

| Methods | Params($M$) | BD-rate |
|---|---|---|
| w/o Atten | 50.34 | -9.08 |
| w/ Atten | 60.48 | -11.64 |
| w/ WDAtten (n=1) | 52.75 | -12.08 |
| w/ WDAtten (n=4) | 60.48 | **-12.93** |
| VTM-17.0 | - | 0 |

**Performance of DREM.** DREM is proposed to dynamically selecting reference subsets of tokens in global range. To illustrate the performance of DREM, we compare our proposed method with different global attention patterns. The global context network is abandoned to build the baseline. The highlight of DREM is adaptivity. Therefore we compare with fixed attention patterns as shown in Table 3. To illustrate the importance of global context informarion, the global context network is discarded. The fully method computes pairwise correlations without attention masks. The Top-K method maintain K most relevant tokens in each prediction and the number K is chosen empirically (Qian et al., 2022a;b), lacking of flexibility.

Table 3: Comparison of different attention patterns. The RD perfermance on Kodak datasets are displayed.

| Methods | BPP | PSNR |
|---------|-----|------|
| w/o $g_{gl}$ | 0.311 | 30.875 |
| Fully | 0.279 | 32.118 |
| Top-K | 0.288 | 31.980 |
| DREM | **0.271** | **32.376** |

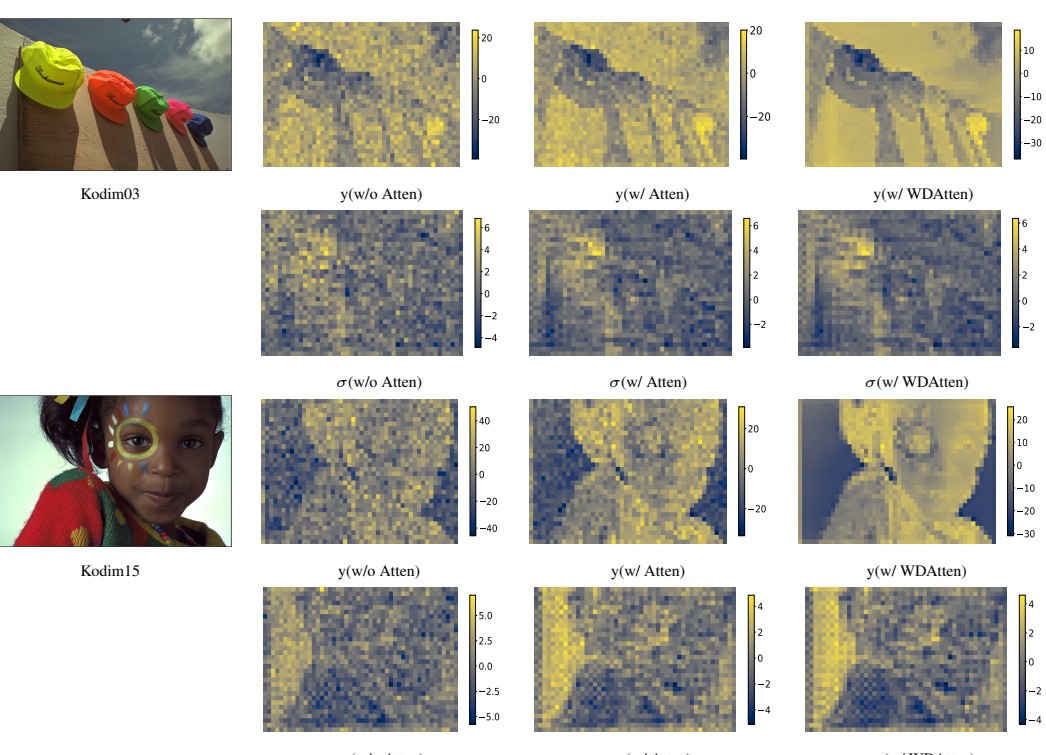

Figure 5: The average scaled deviation $\sigma$ and feature $y$ across channels. The model (w/o Atten) abandons the WDA module and (w/ Atten) and (w/ WDAtten) denote adopt vanilla attention patterns and dynamic sparse attention patterns with the WDA module respectively.

## 5 CONCLUSION

In this paper we first adopt dynamic attention into learned image compression. Based on the assumption that the redundancy information densely distributes in local regions and sparsely exists in long-range distance, we propose the WDA module to dynamically sparsifying the attention matrix in Swin-T blocks, making adaptive attention patterns learned from data possible. This is reasonable because of the diversity of image entropy distribution. The WDA module dynamically modulate the contextual information according to the local entropy, where regions with large entropy could be allocated more long-range context. Appling the WDA into the entropy model, the proposed dynamic-reference entropy model select a subset of reference tokens, sparsifing the optimization space and decreases the risk of overfitting. Extensive experiments demonstrate the performance advantage of our method and proves the possibility for compression networks to evolve in a dynamic and flexible direction in the future.

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

## A APPENDIX

### A.1 DETAILS OF THE ARCHITECTURES

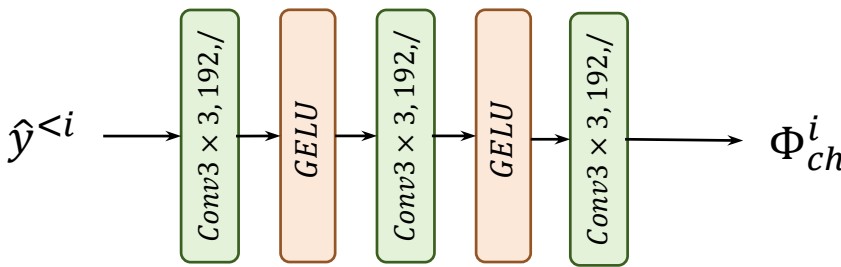

Figure 6: The architecture of $g_c h$.

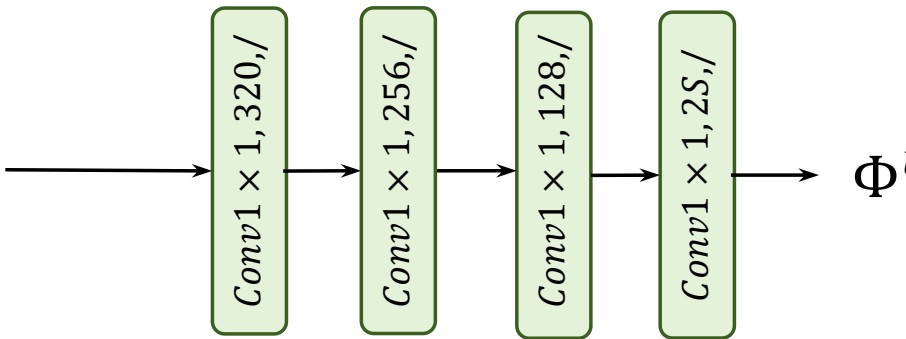

Figure 7: The architecture of $g_e n$.

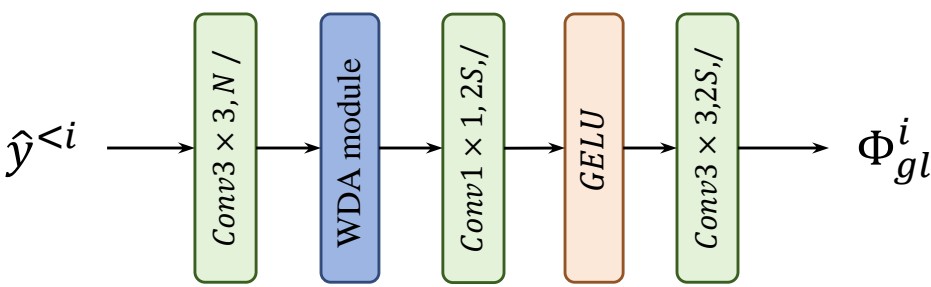

Figure 8: The architecture of $g_g l$.

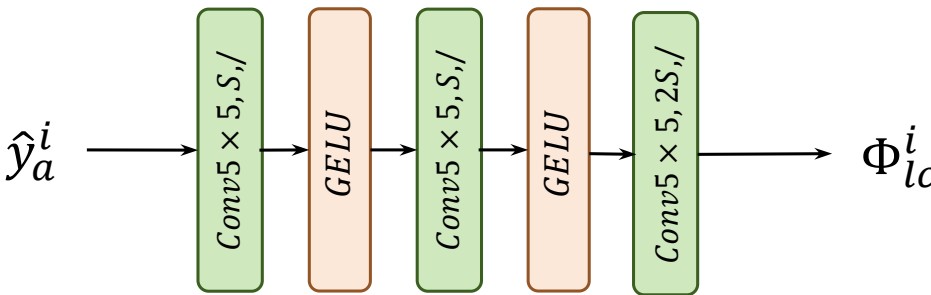

Figure 9: The architecture of $g_l c$.

## A.2 MORE RD PERFORMANCE RESULTS

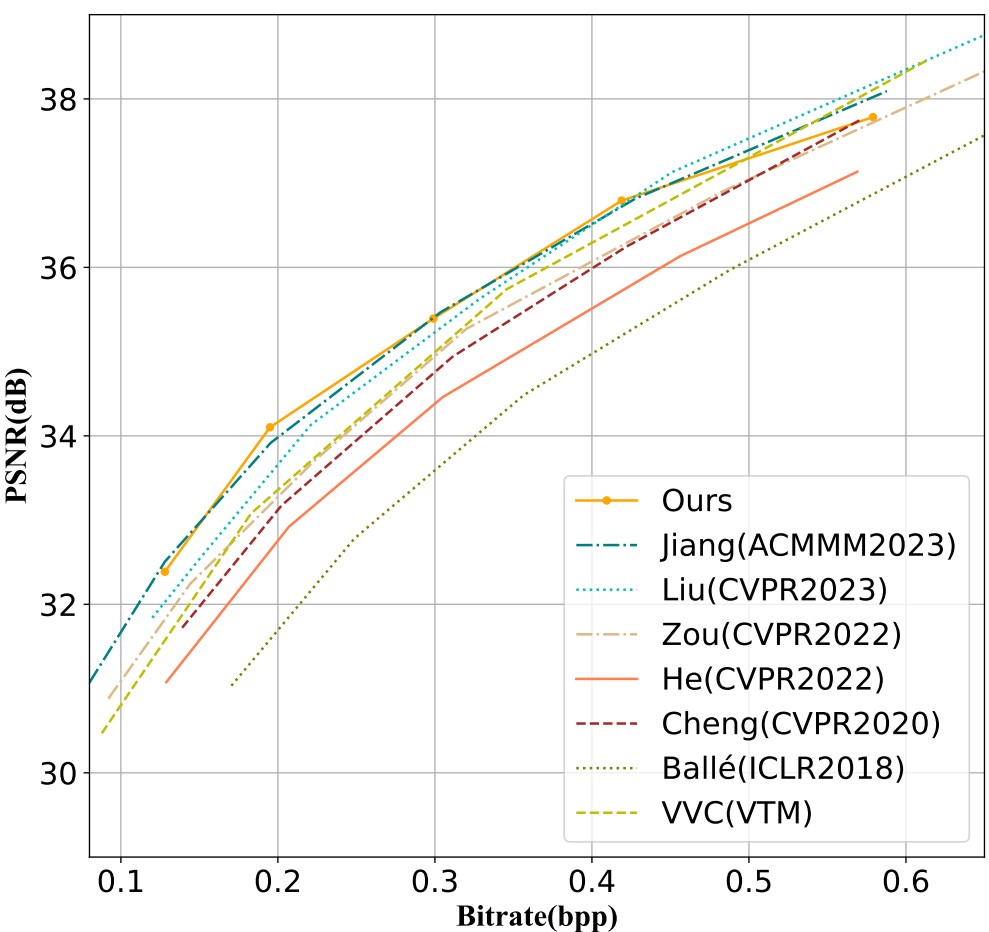

Figure 10: RD performance results on clic professional valid dataset.

702
703
704
705
706
707
708
709
710
711
712
713
714
715
716
717
718
719
720
721
722
723
724
725
726
727
728
729
730
731
732
733
734
735
736
737
738
739
740
741
742
743
744
745
746
747
748
749
750
751
752
753
754
755

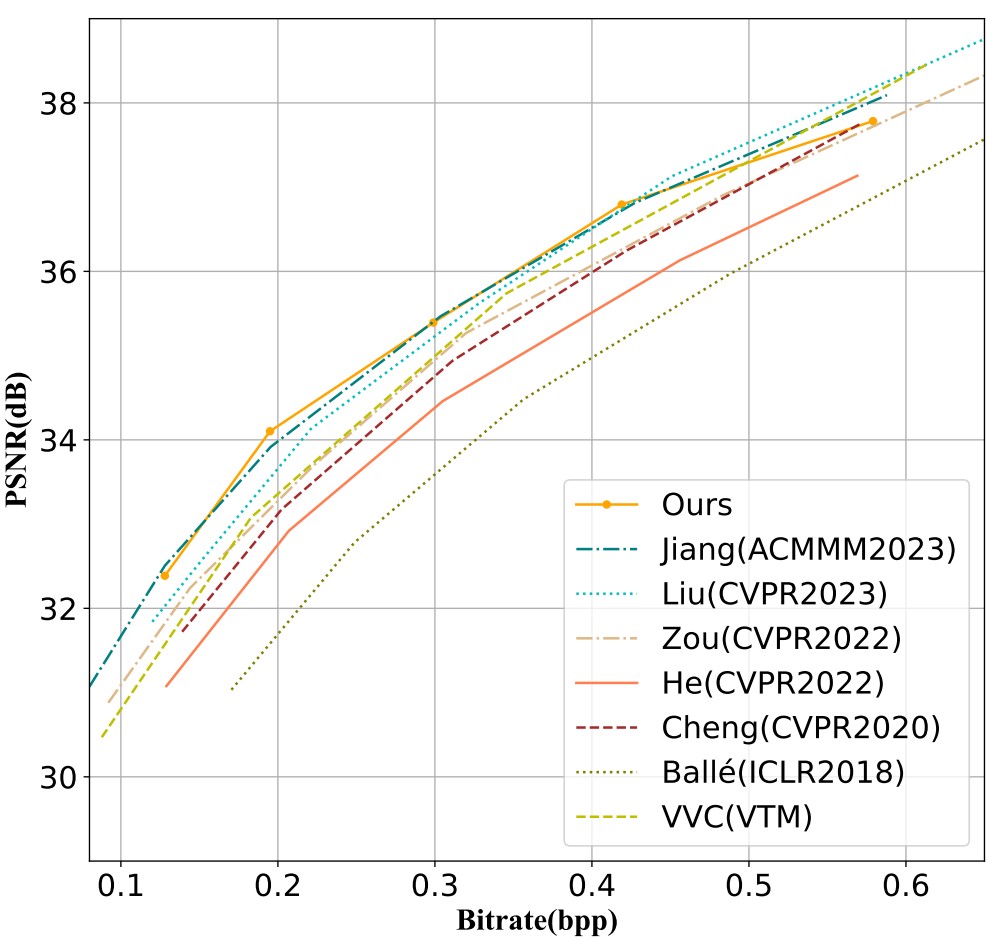

Figure 11: RD performance results on the Tecnick dataset.

