# OpenReview forum: "Window-Based Hierarchical Dynamic Attention for Learned Image Compression"
_ICLR.cc/2025/Conference — Submitted to ICLR 2025_

### Official Review · Reviewer_1huT · 2024-10-23

**Soundness:** 2
**Presentation:** 2
**Contribution:** 2
**Rating:** 3
**Confidence:** 5

**Summary:**

This paper introduces a dynamic attention mechanism to learned image compression (LIC), motivated by the observation that referencing irrelevant content can mislead probability estimations and lead to overfitting.

**Strengths:**

The paper presents an interesting perspective by addressing attention in image compression, highlighting the inherent redundancy in vision transformers (ViT).

**Weaknesses:**

1. The use of a dynamic attention mechanism, while relevant, has been extensively explored in the literature. Therefore, introducing it to the LIC architecture does not constitute a significant contribution. It is suggested that the paper should emphasize the difference with related works in Sec. 2.2 about network architecture, and the issues when applying current dynamic attention modules to LIC. The paper should have delved deeper into the underlying reasons for redundancy in ViT (e.g., proving the overfitting in ViT-based LICs through experiments showing testing error curves). The only difference of proposed Dynamic-Reference Entropy Model (DREM) is adding dynamic attention module.

2. The performance gain is quite marginal, showing even degraded performance on Tecnick and CLIC datasets. For example, the PSNR is lower than VVC and Jiang (ACMMM2023) in Tecnick and CLIC.

**Questions:**

1.	Why does the rate-distortion (RD) performance on the Tecnick and CLIC datasets show an obvious drop?
2.	For a fair comparison, the paper should include results against state-of-the-art dynamic attention works (e.g., the works mentioned in Sec. 2.2), which can easily be adapted to LIC by swapping out modules.
3.	The encoding/decoding complexity of the proposed model should be compared with baseline models to evaluate the impact of the dynamic attention mechanism on computational complexity.
4.	The paper contains grammar and spelling issues, such as lines 288 and 291, which should be addressed.

---

### Official Review · Reviewer_nRaL · 2024-10-30

**Soundness:** 2
**Presentation:** 3
**Contribution:** 1
**Rating:** 3
**Confidence:** 5

**Summary:**

This paper proposes a window-based dynamic attention module (WDA) that adapts attention patterns to reduce redundancy in global contextual information. The core idea is to compute a covariance matrix, which sparsifies the attention weight matrix based on correlations. The WDA module is integrated with an advanced framework to develop a fairly effective learned image compression (LIC) algorithm.

**Strengths:**

1. The paper introduces a sparsified attention mechanism that leverages covariance matrices to adjust attention weights at a fine-grained level based on feature correlations.
2. The method achieves decent results on the Kodak dataset.

**Weaknesses:**

1. The novelty of the paper is limited, focusing mainly on the introduction of a new attention module, the window-based dynamic attention (WDA) module. While the module demonstrates some performance gains in experiments, the contribution lies largely in refining existing Transformer structures rather than introducing new frameworks or theories.
2. Although WDA and the dynamic-reference entropy model (DREM) improve compression performance, they also increase computational overhead. This additional complexity could make the approach impractical, especially when processing high-resolution images, as the dynamic attention mechanism requires significant computational resources.
3. While the paper showcases the performance advantages of WDA and DREM, it lacks detailed analysis regarding the impact on complexity, computational cost, and decoding latency. These aspects are critical for real-world applications, and the absence of such evaluations makes it difficult to assess the model's practical value and feasibility for deployment.

**Questions:**

1. Why does the method perform poorly at high bitrates on the CLIC and Tecnick datasets? This inconsistency with the results on Kodak is puzzling, especially since the results on CLIC and Tecnick align with each other. How do the authors explain this discrepancy?
2. How much additional computation and parameter overhead does the introduction of covariance calculations bring?
3. Does the method increase decoding latency?

---

### Official Review · Reviewer_roFj · 2024-10-31

**Soundness:** 3
**Presentation:** 1
**Contribution:** 2
**Rating:** 5
**Confidence:** 4

**Summary:**

This paper proposes a window-based dynamic attention (WDA) module to improve learned image compression (LIC) by addressing overfitting issues in Vision Transformer (ViT)-based models. Unlike traditional methods that rely on fixed attention patterns, the WDA module dynamically adjusts attention patterns within local windows based on entropy information, focusing only on relevant contextual features. Additionally, a dynamic-reference entropy model (DREM) is introduced to enhance probability estimation by adaptively selecting informative decoded symbols.

**Strengths:**

The research is thorough, with rigorous mathematical formulations and comprehensive experiments across multiple datasets. Core ideas and methodology are clearly presented, though minor improvements in terminology and figure alignment could enhance clarity. By addressing overfitting in ViT-based LIC, the paper offers valuable insights for the field of transformer-based image compression, with demonstrated gains in compression efficiency that could impact future applications.

**Weaknesses:**

a. Lack of clarity of motivation: The relationship between long-range modeling and overfitting is inadequately explained. The passage suggests that ViT's ability to capture distant context may lead to overfitting, but it lacks a clear connection between these two factors in the context of learned image compression.
b. The experimental comparisons rely on outdated methods, lacking evaluations against more recent and advanced techniques [1,2,3].
c. The paper suffers from vague terminology and unclear references, such as the undefined use of terms like "the sequence" in L215.
d. Fig.2 contains inaccuracies, such as incorrectly depicting 𝑄 and 𝐾 as square matrices instead of 𝑁×𝑑𝑘 matrices. Two 4*4 matrices cannot output 16*16 matric through matrix multiplication operation.
e. The paper lacks a comparison with more advanced masking techniques [4,5]. As the authors mentioned, the fixed Top-K attention can also bring RD performance gains in L240-242.

1. FTIC: Freguency-Aware Transformer for Learned Image Compression, ICLR 2024.
2. GroupedMixer: An Entropy Model with Group-wise Token-Mixers for Learned Image Compression, TCSVT 2024.
3. Causal Context Adjustment Loss for Learned Image Compression, NIPS 2024.
4. EulerMormer: Robust Eulerian Motion Magnification via Dynamic Filtering within Transformer, AAAI 2024.
5. Entroformer: A transformer-based entropy model for learned image compression, ICLR 2022.

**Questions:**

There is no question at this time.

---

### Official Review · Reviewer_eXFH · 2024-11-02

**Soundness:** 2
**Presentation:** 3
**Contribution:** 2
**Rating:** 3
**Confidence:** 5

**Summary:**

This paper proposes a Window-Based Hierarchical Dynamic Attention Learned Image Compression (WDA-LIC) method. It uses the WDA module to sparsify attention matrices based on entropy, adaptively learning attention patterns to solve challenges like overfitting and inaccurate entropy estimation.

**Strengths:**

This paper is well organized and is clear and easy to understand.

**Weaknesses:**

1. Only the methods in 2023 and before were compared in the paper. It is necessary to make a comparison with [1] in 2024.
2. There are relatively few innovation points. As stated in Section 2.2, the dynamic attention is a method that already exists in other fields. The author just applied it to image compression.
3. Regarding the first point of the innovation points, some previous works, such as [2] and [3], have already explored it.
[1] Frequency-Aware Transformer for Learned Image Compression. H Li et al.
[2] Learned Image Compression with Mixed Transformer-CNN Architectures. J Liu et al.
[3] Checkerboard Context Model for Efficient Learned Image Compression. D He et al.

**Questions:**

See weakness. Please clarify the contributions.

---

### Official Review · Reviewer_qZbo · 2024-11-03

**Soundness:** 3
**Presentation:** 2
**Contribution:** 2
**Rating:** 3
**Confidence:** 4

**Summary:**

The paper studies the redundancy problem of learned image compression and develops two dynamic attention modules for this problem (based on multiscale and directional analysis). The method introduces these two modules to latent transformation network and entropy model respectively. Based on WDA and DREM, the learned image compression achieves better rate-distortion performance. The method shows improvement over learned codec and conventional codec baselines by a healthy margin.

**Strengths:**

1. The proposed WDA and DREM dynamic attention modules capture redundancy in latent space
2. This method leads to consistent rate–distortion performance improvement across diverse learned image compression benchmarks.

**Weaknesses:**

1. My major concern is the limited technical novelty and contribution of the paper. Dynamic attention is a simple idea but just a variant of the attention -- use covariance matrix to sparsify the attention matrix. It compensates for the top-k method.
2. As far as I know, it is a challenge to apply transformer to image compression. Window-based attention somehow eases the overfitting problem. The authors are suggested to construct more analysis on the motivation of applying dynamic for window-based attention.
3. It is interesting to find that dynamic attention achieves a significant improvement compared to the non-dynamic method. However, it is not clear that how does the threshold $t$ outcome. The authors are suggested to provide an ablation study on threshold.

**Questions:**

1. More analysis on the motivation of applying dynamic for window-based attention as W2.
2. In Sec 4.3, "Atten denotes plain attention patterns that discards masks", does it denote plain window-based attention, or full attention across all pixels?
3. Section 3.3, which discusses DREM and Equation 12, requires reorganization to enhance clarity and coherence.
4. Some typos. Line 193, the Figure reference is missed. Line 284, the Figure reference is missed.

---

### Meta-Review · Area_Chair_Bjkt · 2024-12-18

**Metareview:**

This paper received all negative ratings from the reviewers. All the reviewers have raised concerns on the novelty of the proposed work and the motivations. At the same time, the authors did not provide a rebuttal to reviewers' comments. Therefore, AC made decisions based on the reviewers' recommendations.

**Additional Comments On Reviewer Discussion:**

This paper received all negative ratings from the reviewers. The authors did not provide a rebuttal to reviewers' comments.

---

### Decision · Program_Chairs · 2025-01-22

Reject